# Building a Research Roadmap for Caregiver Innovation: Findings from a Multi-Stakeholder Consultation and Evaluation

**DOI:** 10.3390/ijerph182312291

**Published:** 2021-11-23

**Authors:** Kieren J. Egan, Kathryn A. McMillan, Marilyn Lennon, Lisa McCann, Roma Maguire

**Affiliations:** 1Digital Health and Wellness Research Group, Department of Computer and Information Science, Faculty of Science, University of Strathclyde, Glasgow G1 1XH, UK; katie.mcmillan3@nhs.scot (K.A.M.); marilyn.lennon@strath.ac.uk (M.L.); lisa.mccann@strath.ac.uk (L.M.); roma.maguire@strath.ac.uk (R.M.); 2Information and Digital Technology Department, NHS Shetland, Lerwick, Shetland ZE1 0LB, UK

**Keywords:** caregivers, innovation, research, co-design, interdisciplinary, digital health, participatory design, collaboration at distance

## Abstract

Across the world, informal (unpaid) caregiving has become the predominant model for community care: in the UK alone, there are an estimated 6.5 million caregivers supporting family members and friends on a regular basis, saving health and social care services approximately £132 billion per year. Despite our collective reliance on this group (particularly during the COVID-19 pandemic), quality of life for caregivers is often poor and there is an urgent need for disruptive innovations. The aim of this study was to explore what a future roadmap for innovation could look like through a multi-stakeholder consultation and evaluation. An online survey was developed and distributed through convenience sampling, targeting both the informal caregiver and professionals/innovators interested in the caregiver demographic. Data were analysed using both quantitative (summary statistics) and qualitative (inductive thematic analysis) methods in order to develop recommendations for future multi-stakeholder collaboration and meaningful innovation. The survey collected 174 responses from 112 informal caregivers and 62 professionals/innovators. Responses across these stakeholder groups identified that there is currently a missed opportunity to harness the value of the voice of the caregiver demographic. Although time and accessibility issues are considerable barriers to engagement with this stakeholder group, respondents were clear that regular contributions, ideally no more than 20 to 30 min a month could provide a realistic route for input, particularly through online approaches supported by community-based events. In conclusion, the landscape of digital health and wellness is becoming ever more sophisticated, where both industrial and academic innovators could establish new routes to identify, reach, inform, signpost, intervene and support vital and vulnerable groups such as the caregiver demographic. Here, the findings from a consultation with caregivers and professionals interested in informal caring are presented to help design the first stages of a roadmap through identifying priorities and actions that could help accelerate future research and policy that will lead to meaningful and innovative solutions.

## 1. Introduction

Informal (unpaid) carers, also termed “caregivers”, are family members and friends who support a loved one who needs help due to illness, frailty, disability, mental health problems or addiction. In the UK, it is estimated that there are at least 6.5 million informal caregivers, a workforce substantially larger than the National Health Service (NHS) [1] and the collective saving to the health and social care services is estimated to be £132 billion per year. This situation is similar across Europe, and beyond, where 80% of all care is delivered by informal caregivers [2]. As the tide of an ageing global population continues to advance in tandem with a shrinking health and social care workforce [3], there is a public health emergency looming, whereby the pressures on caregivers across the world are set to significantly escalate.

While the recent events during the COVID-19 pandemic have demonstrated the societal value of caring, it also highlighted that many will face short, and long-term, health and wellness consequences. Informal caregivers share many of the same challenges of a professional workforce, such as NHS employees, but lack much of the associated infrastructure. For example, there is little to no: support, training, pre-agreed workload such as hours per week or a working role definition, and caregivers often must balance caring responsibilities around work, families and their own health and wellness needs. While some caregivers do enjoy benefits from caring roles, such as a sense of fulfilment from caring [4], there is now compelling evidence that caregiving adversely impacts on health and wellness, both in the short and long term [5,6,7]. Crisis points, such as hospitalizations, significant worsening of mental and/or physical health and irreversible changes to caring circumstances are commonplace (even in the absence of COVID-19) and frequently cause deterioration in health for caregivers, and those being cared for [8,9].

Given the considerable number of unmet needs described above, there have been many different innovations developed for the caregiver demographic. Although successful evidence has emerged [10,11,12], there still remains a striking number of caregivers who remain “out of reach” [8] through a combination of factors, including (but not limited to): digital and health literacy levels, socioeconomic status, health, mobility, and level of dependency required from the person cared for. Thus, there remains a continued need to consult, innovate and evaluate for, and with, caregivers and to develop research excellence in this area. Emerging from the evidence in this field, there are several areas of particular interest, which include: methodologies to improve identification of caregivers (including in hard to reach groups such as ethnic minorities [13], the development and adaption of theoretical/conceptual frameworks, techniques for cultural adaptation [14] and addressing implementation and abandonment issues [15].

Digital health and wellness solutions are a core part of the current drive to deliver equity in health to all—including informal caregivers. In 2020, for example, the WHO (World Health Organization) produced a long-term strategy for the use and scale-up of digital health, highlighting the positive impact digital health can have on the access to, and provision of healthcare as well as the health and wellbeing of the population [16]. Such findings are paralleled in recent review work [17]. Much of this work focuses on caregiver health and wellbeing or signposting to sources of support and are not restricted to one or two types of technology (e.g., mobile app, internet-based, integrate platforms, sensors) [17]. Despite such progress, the market reality is that most startups fail and many successful research studies are challenging to implement. Reducing uncertainties and increasing the availability of market knowledge, such as bettering current understanding of ongoing caregiver needs, would be considerably advantageous.

Taken collectively, innovation for informal caregivers is sorely needed but the landscape still falls short of meeting the everyday needs. Identifying both better ways to collaborate across all stakeholders and methodologies to rapidly, but robustly, develop and test innovations could help improve the translational hit rate. Therefore, here we took a first step to improve engagement with caregivers by undertaking a multi-stakeholder consultation to design a future roadmap for innovation in caregiver research.

## 2. Materials and Methods

Initially, this study was due to involve face-to-face interviews and focus groups, however, due to the COVID-19 pandemic we had to be agile and to achieve our aim of engaging with as many caregivers and other stakeholders as we could, we changed methodology and conducted the multi-stakeholder consultation via survey. We developed our own survey approach as we did not find a suitable existing one. Although the same content of surveys was delivered to both caregivers and professional groups (e.g., healthcare professionals, innovators), we tailored wording on occasion to represent each stakeholder group.

The surveys were iteratively designed to collate participant information regarding: (i) demographics, (ii) previous research involvement, (iii) feelings towards research and innovation, (iv) barriers to taking part in research, (v) future participation in research and, (vi) focus/outcomes of future research. No questions were mandatory and, thus, the total number of responses for each question varied.

Ethical approval was granted from the Computer and Information Science (CIS) Department Ethics Committee at the University of Strathclyde, Glasgow. The surveys, aimed at caregivers and those with a professional role interested in caregivers, were designed to take no more than 10 min to complete, and to be used across the UK. See Appendix A for example questions.

Inclusion criteria and survey design: Participants 18 years and over were invited to take part. A broad definition of an informal (unpaid) caregiver was shared with participants, “*People that provide unpaid care by looking after an ill, older or disabled family member, friend or partner*”: hereafter referred to as caregivers. A minimum number of hours per week that caregivers needed to be caring for was not specified, leaving it up to participants to self- identify with the term. For professional groups, our participant information sheet stated that we were interested to find “*Professionals (not employed by Universities) who have an interest in innovating around carer health and wellbeing (e.g., health and social care professionals or those working in the technology industry/third sector)*”. Where we state professionals, hereafter, this implies professionals/innovators interested in the caregiver demographic.

Survey distribution: Consent was implied after participants read, acknowledged, and accepted the initial terms of the anonymised survey. The distribution of the survey involved sharing the online version of our questionnaire through social media channels (e.g., Twitter), alongside email distribution through networks accessible to partners, such as the Digital Health & Care Innovation Centre (DHI) and Carers UK (e.g., Healthy Ageing Innovation Group and the “Digital Health & Care” Mailing list). The survey was open from 15 June 2020 until 30 September 2020.

Data handling: The survey was constructed using Qualtrics Software. Participants were reminded that any data entered must not contain any identifiable information. An integrated mixed-methods approach was taken for the data analysis. All quantitative analyses (frequencies and summary statistics) were completed using R studio (version 1.1.456). Qualitative analyses were undertaken using a content analysis approach by two researchers (KM and KE) [18]. Free text responses to questions were collated and the first question coded by KM. The coding structure cross-checked and agreed by KE as a measure of inter-rater reliability. Once agreed, the rest of the coding was completed by KM, before being cross-checked by KE. Data were frequently referred back to throughout this process to ensure the coding framework developed was appropriate. The flexible nature of content analysis tolerates the use of a combination of deductive and inductive creation of themes and patterns from the data. Deductive analysis was used to create the key themes and inductive analysis was used to create sub-themes. More specifically, the six key themes were identified from the survey structure (deductive analysis), sub-themes were identified through researcher (KM and KE) interpretation and coding of free text responses (inductive analysis). Data gathered from informal caregivers and professionals were analysed, and are presented, separately in the results section of this paper. Major themes were identified where at least half of respondents aligned with a specific finding. Where respondents identified as both a caregiver and a professional/innovator interested in caregiving, we classed these individuals as part of the caregiver demographic.

Data analysis: This work was developed without an a priori hypothesis but instead was conducted as an observational/explorative piece used to generate a future roadmap/hypothesis. Therefore, our statistical analysis is limited to frequencies, presented as stacked bars throughout the manuscript (we interpreted priorities where >50% of the respondents suggested an item was important/very important or equivalent). Qualitative analysis included the identification and discussion of key themes, such as clear priorities outlined by participants, throughout the narrative. For the purposes of readability and figure simplicity, long statements have been abbreviated and a full list of statements presented to participants can be found in Appendix B.

## 3. Results

### 3.1. Description of Our Sample

#### 3.1.1. Caregiver Demographics

The survey received 112 responses from informal caregivers (see Table 1). Caregiver respondents were aged 25 to up to 75 to 84 years old, with the mode age within the 45 to 54 years category. Over 85% (96/112) of caregiver respondents identified as a woman/female. Caregivers were highly educated, 60.7% were educated to degree level or equivalent, and 111/112 identified as white (Table 1). Just over half (50.9%, 57/112) of caregivers had been caring for 10 or more years and 6% (7/112) had been caring for 1 year or less. Caregivers varied in their ability to undertake work/study alongside their caring role: 35% were able to continue to work/study, 34% had been forced to reduce their work/study hours and 19.6% had been forced to give up work/study. Almost all (96%) of caregiver respondents were using digital technologies, such as smartphones and laptops on a daily basis. The vast majority of caregiver responses were from Scotland (96%, 108/112), with 1 response from Northern Ireland (<1%) and 3% (3/112 responses) from England.

#### 3.1.2. Professionals Demographics

The survey received 62 responses from professionals (Table 1). Professional respondents were aged from 25 to up to 55 to 64 years, most identified as women/female (79%) and 94% of respondents were white. The professional respondents were highly educated, 74% had a degree or equivalent level of education. When asked what their professional role was, 61 of the 62 participants responded. Most respondents (36%, 22/61) worked in healthcare, followed by 25% (15/61) working in social care, 23% (14/61) working in the 3rd sector, 11% of respondents put other, 3% worked in caregiver policy development and one respondent worked in a Small to Medium Enterprise (SME). All participants reported everyday use of technology. Similar to caregivers, 98% (60 /61) of professionals were from Scotland, with 1 response from England (2%).

### 3.2. Views of Informal Caregivers

Three key themes were established from the structure of our questioning and eight sub-themes were identified from the thematic analysis of the professionals free-text survey data. The three pre-specified (deductive) themes were; (1) Previous research participation, (2) Future research participation and (3) Future research aspirations. Sub-themes (identified through inductive thematic analysis) are presented in Table 2.

#### 3.2.1. Previous Caregiver Research Participation

Most respondents (79%, 89/112) had not previously participated in research being conducted by a university in relation to their role as a caregiver, while 18% (20/112) of participants had previously taken part in research. For those who had previously taken part in research, the majority had completed surveys: there was no reference by any of the caregivers to taking part in other forms of research, such as focus groups or interviews. Most had taken part previously due to a professional interest in research: for example, it being part of their job, or from a personal interest whereby participating in the research enabled access to specific support or services. The main reasons for not participating in research previously was lack of awareness, being unaware of any ongoing research or not being asked to participate in research prior to the current project. The other barrier towards participating in research was the lack of free time caregivers had allowing them to commit, given the other time pressures in their life. Time was, by far (both in Likert and free text responses) the most significant barrier to taking part in research, followed by money and personal health (See Figure 1).

Closer inspection of our Likert responses around current and previous experiences of caregiver research suggests over 75% of caregivers agree/strongly agree that caregivers have something “useful to share” with researchers, yet far fewer respondents felt that researchers could make a significant impact on their health and wellness as a caregiver (Figure 1). Few caregivers responding to this work felt they were connected to researchers in universities. When asked to rate whether they felt connected to university research, 29% (24/84) of participants strongly disagreed with this statement (Figure 1). It was suggested that university researchers do not have a particularly strong grasp of the realities of a caring situation.

#### 3.2.2. Future Research Participation

Online and community-based events, along with newsletters, were the most common ways in which caregivers felt they could contribute to university-led research in the future. However, when asked what the best way would be for caregivers to work alongside academia in the future, online and in person were the most popular options, and almost equally split among participants. The split opinion seems to be between the gold standard being face-to-face to ensure a rapport is built between caregiver and researcher, among other benefits, and the practicality and perhaps reality that online would be more suitable, given the time pressures faced by caregivers. The majority (53%, 39/73) of respondents would be willing to commit 30 min or more a month to participating in research.

#### 3.2.3. Future Research Aspirations

In terms of future focus for the health and wellbeing of caregivers, from Likert scale responses, there was a need stated for a diverse range of areas including: information-based research, remote monitoring technology, communication technologies to enable connection between caregivers and policy related research to help shape national agendas and policies (Figure 2a). When asked about different ways in which informal caregivers could provide input in research, many placed high value on all the options presented to them, including developing new ideas and new approaches to working together. Other popular options included ensuring that fresh solutions are relevant for other caregivers. Reviewing innovative technology ideas in development was not a clear priority for caregivers compared to other solutions. Thematic analysis of participant quotations highlighted a strong need to improve the support provided to caregivers. This included financial, emotional, psychological and training and educational support. It was very apparent that participants do not currently feel heard or that they have received sufficient levels and/or quality of support in their role as a caregiver.

Further feedback for future work was obtained around which specific areas of collaboration would make the biggest impact on caregiver health and wellness (Figure 2b). There was a wide array of different interests in this group, for example, information science was a particular area of interest highlighted where caregivers viewed research that shares key information around: caring policy, rights, and entitlements as a priority. While more than 50% of the participants viewed research around mobility, innovation and learning, finance and voice activated research as useful/very useful, fewer participants were interested in Virtual Reality/Augmented Reality as a research priority.

### 3.3. Professionals

Three key themes were established from the structure of our questioning and eight sub-themes were identified from the thematic analysis. The three pre-specific (deductive) themes were; (1) Previous research participation, (2) Future research participation and (3) Future research aspirations. The key themes and sub-themes are presented in Table 3.

#### 3.3.1. Previous Research Participation

Professionals were asked if they had previously been involved with or collaborated with universities and just over 62% (38/61) of respondents said they had not. When asked to comment further, many of those who had previous experience with research had done so in a professional capacity as part of their role. A key reason for having participated in research or being interested in doing so in the future was the importance and want to share their knowledge and experience.

The majority of professionals suggested that caregivers do not feel connected to universities or researchers and, similarly, many did not think that those working in universities have a good understanding of the challenges faced by caregivers (Figure 3). However, the vast majority of respondents felt that caregiver experiences would be extremely valuable to researchers and that research could have a significant impact on the health and wellbeing of caregivers (Figure 3). When asked about barriers to caregivers participating in research, lack of time was the most significant barrier, followed by money and then poor caregiver health.

#### 3.3.2. Future Research Participation

Professionals stated that online group activities and community-based events were the most popular ways which would make it easier or more likely for people to contribute to caregiver research 82% (46/56) and 66% (37/56), respectively. However, when asked to explain further, the importance of engaging those who are harder to reach and not simply using charity organisations as a means of recruiting caregivers to participate in research came through strongly in the free text answers. Perhaps, reflective of the COVID-19 situation people were living in at the time of completing the survey, online methods followed by in person face-to-face meetings were preferred with phone-based and postal methods being less popular. Many respondents, however, highlighted the benefits of face-to-face meetings.

#### 3.3.3. Future Research Aspirations

We asked professional respondents similar questions to caregivers around research priority areas. Although there was some indication that finding available solutions and measuring success were important, training and support, planning ahead, developing new ideas and developing new approaches to work with research were all a priority that over 50% of professionals thought were “important” or “very important” (Figure 4a). In terms of specific types of technology, respondents felt research and development into all areas of technology would be useful (Figure 4b). Those respondents appeared to think that what would be particularly useful were technologies enabling remote monitoring and communication technologies to facilitate connection between caregivers and other caregivers or professionals. Interestingly, virtual reality technology was of interest, but noticeably less so than other forms of technology (Figure 4b).

When asked to elaborate on the most important outputs they felt could come from their involvement in caregiver research, the response fits into four key themes: (1) Solutions that are relevant to caregivers and truly reflective of their needs, (2) Long-term, sustainable solutions, (3) For caregivers to feel empowered and supported in their role, (4) Possible impact on policy and practice.

## 4. Discussion

The pressures on the caregiver demographic continue to grow. An ageing growing global population [19], the disruption caused by COVID-19 [8] and limited support for the caregivers are a perfect storm for a population already under strain. Recent UK statistics suggest that year on year, that stress outcomes have been worsening for caregivers, underlining the need for disruptive change [20]. Here, we have explored what a future roadmap for innovation could look like through a multi-stakeholder consultation and evaluation: an area of considerable importance for public health. Through incorporating the views of 174 individuals (including 112 informal caregivers and 62 professionals), there are a wide range of unmet needs, and an appetite to work closer with researchers and universities through both in person and online approaches (see Figure 5). While our responses suggest that working together to understand core needs is plausible for a sizable group of caregivers, care must be taken not to become overzealous in our interpretations—many caregivers may remain difficult to reach, including those who are delivering care on 24/7 basis. Nevertheless, this work forms some early steps to engaging the wider caregiver demographic and, taken collectively, our findings suggest a need to: (i) identify and work sustainably with caregivers, (ii) listen to, and encompass the wide range of needs and perspectives of caregivers and other stakeholder groups, (iii) improve the quality and quantity of methodologies for caregiver research and (iv) widely share research knowledge (e.g., successful and non-successful innovations for research/implementation).

While we achieved our main objectives, there are several limitations. Our work represents an online survey where there are biases caused by convenience sampling. For example, representation from ethnic minorities is relatively sparse, however, this was not a specific objective we aimed to address and further work is required to understand whether our findings are paralleled in such groups. We may also be inadvertently missing out on views on those with substantial caring commitments or those who do not engage in digital technologies. Second is that, while we have attempted to be as objective as possible in our analyses and interpretations, our questions were pre-specified and, therefore, our qualitative analysis should be interpreted as deductive in terms of thematic approach. Third is that there are many important barriers to innovations that we cannot comment on, such as government incentives, cost-benefit models and start up environments. Our responses are based almost exclusively in the Scottish setting: in future, it would be advantageous to better understand other parts of the UK, alongside more nuanced differences such as differences between urban vs. rural settings. Finally, this work took place during the COVID-19 pandemic, which may have skewed representation to more digitally advanced groups of caregivers and missed important viewpoints.

Our main findings highlight a need to identify and work sustainably with caregivers: a task not straightforward given the lack of emotional, physical, and monetary support readily available. Moving forward, there is a need to find better ways to recognize the value that caregivers bring. This may be achievable through developing a more continuous interaction with caregivers, for example, moving beyond individual projects to open ended or problem focused collaborations. While many caregivers would welcome routes to engage with universities, researchers must see that the process needs to be efficient and inclusive: ideally taking no more than 20 to 30 min a month. Relevant to such a trade-off between researchers aims and citizen engagement is the concept of living labs. Living labs are a methodology/approach that can allow citizens to participate in the design, development, and evaluation of innovative solutions to address societal problems [21]. There is considerable interest in using such approaches to explore health and wellness problems, including scope to carefully explore a range of different issues through participatory design, across the life course [22,23]. These approaches are helping to redefine research methodologies, including the integration of citizen generated and non-traditional data sources [24,25]. Although careful consideration will be required of how to best build upon emerging online methodologies (e.g., ethical use of data), our findings here indicate that a hybrid model of both online and in-person approaches could work well for the caregiver demographic. “Citizen science” approaches could facilitate multiple research stages, such as ideation, co-design of the approach, data-gathering and knowledge transfer of findings through “light touch” interactions (e.g., smartphone/computer-based) or through community “pop up” events. Critically, this would allow multiple routes to help different groups of caregivers influence research agendas within the earliest stages [26]. Such integration could help address known issues around translational failure [15] and be supported through a wide variety of tools such as prioritization methodology [27] alongside recognition in academic publications [28].

Another key finding from this work is that caregivers (who were able to respond to this work) would largely welcome being “championed” and provided with a channel to engage with universities. However, the practicalities of delivering this are yet to be realized—political and social landscapes change over time, as do priorities. While both we and many other research groups [29,30] consult caregivers about current unmet needs on a regular basis, there remains a risk that some specific groups (e.g., according to age, geographical location, ethnicity, socioeconomic status, number of hours caring per week) will be excluded from conversations through selection bias [31]. Further work is needed to contextualize how representative current research is of all caregivers and, critically, we must establish community links to help those in need who do not traditionally connect with research agendas or third sector partners. Lastly, while the concept of sustained engagement proposed here may be new to many university settings, there is a need to ensure efforts augment, opposed to replace existing efforts ongoing elsewhere (e.g., health and social care data, charity, and voluntary sector annual reports).

Given that much of the conversation around caregivers, needs and innovations falls out with traditional models of empirical research (i.e., Randomized Controlled Trials), there is a need to keep furthering knowledge of research methodologies—this could be to improve the value of non-traditional data sources and/or to increase capacity for more caregivers to become involved. A source of inspiration that highlights passive involvement from participants includes the “Dreamlab” project from University College London [32], developed so that individuals can engage with researchers through charging their smartphone to assist with cancer research. As the concept has already reached 1 million downloads, this is a clear indication that citizens are interested to connect and make impacts to research priorities. Further, the recent contributions of the COVID ZOE smartphone app have also underlined the power of citizen science, whereby seemingly small actions of individuals on a larger scale can accelerate current knowledge to help navigate the unchartered waters of an emerging pandemic [33]. Given such examples and our participant feedback, it is feasible to see smartphones being better used to connect caregivers (and other citizens/professionals) to ongoing research opportunities and emerging agendas, and to signpost others to recruitment calls. An immediate utility of such citizen science approaches could be to connect caregivers of rarer/under-researched conditions to highlight such needs to researchers and funding councils alike.

Another key take-home point from this work is the demonstration of the real depth of knowledge that caregivers hold for the development of future innovations and co-design approaches [34]. For example, caregivers demonstrated intimate knowledge of the daily barriers and enablers to existing technologies, and from the findings presented here, many caregivers are still actively seeking solutions around information needs and remote monitoring, problems that researchers may think are already solved. In terms of contributing specifically to ongoing research, findings from our own work and others [17,29] have highlighted key research topics for caring (e.g., access to information online), and such priorities could form the foundation of conversations with caregivers in a variety of settings. An achievable and realistic first step from here could be to pick exemplar caregiver priorities, and to run pilot projects that could facilitate the development of caregiver co-designed “principles” to help scaffold the way by which researchers engage with caregivers moving forward.

## 5. Conclusions

To conclude, the aims of this work were to initiate the development of a roadmap for future caregiver innovation. We found that, despite a number of barriers to participation, many caregivers are willing to regularly contribute to the research agenda and have a wide range of unmet needs that could be better addressed by existing and future research outputs. Visibility is still an issue to caregivers—we were only able to hear from a small subsample of the caregiver population here and even these participants did not feel visible and heard in research agendas. As a wider community, this needs to change. Clearly, accessible routes for engagement (e.g., integrated online and face-to-face approaches) and more formal recognition of their expertise would be welcomed alongside the delivery of real-world impacts on those who care. A next logical step would be to determine which approaches (such as living lab methodologies) are capable of delivering for caregivers not just in terms of research outputs/outcomes, but in terms of transparency, engagement, and experience.

## Figures and Tables

**Figure 1 ijerph-18-12291-f001:**
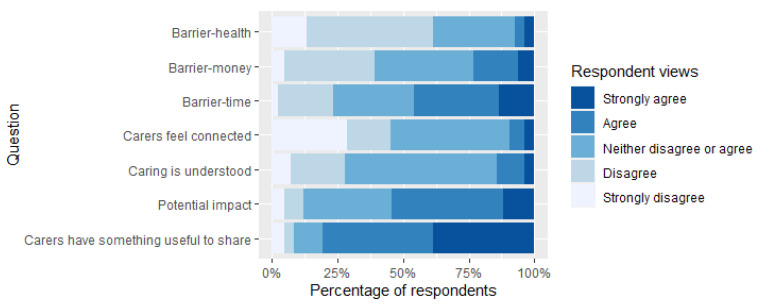
Caregiver responses to a range of statements according to whether they “Strongly agree”, “Agree”, “Neither disagree or agree”, “Disagree” and “Strongly Agree”. Statements focused on whether: (i) caregivers have something useful to share with researchers; (ii) researchers could make potential impact on caregiver health and wellbeing, (iii) the role of a caregiver is understood, and (iv) whether caregivers feel connected to researchers at university settings. In addition, we explored views on the barriers of (v) time, (vi) money and (vii) health to participation in research.

**Figure 2 ijerph-18-12291-f002:**
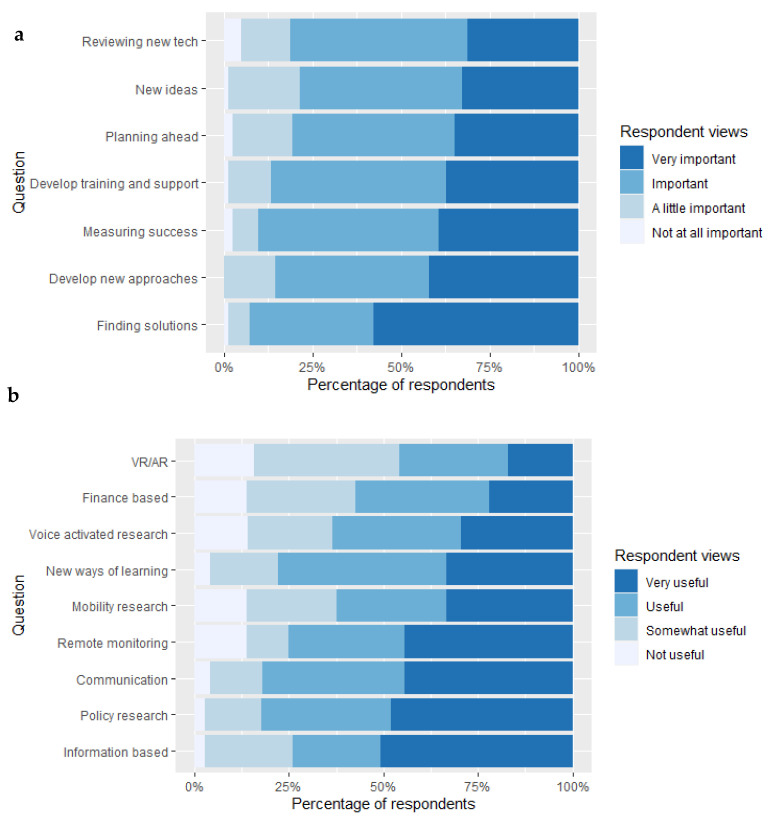
(**a**) Caregivers’ responses to a range of statements according to whether they are “Very important”, “Important”, “A little important”, “Not at all important”. Statements included whether caregivers were interested in (i) finding solutions, (ii) developing new approaches to work with researchers, (iii) measuring success, (iv) developing training and support, (v) planning ahead, (vi) developing new ideas, and (vii) reviewing new technologies. (**b**) A separate question for caregivers focused on whether future work would be “Very useful”, “Useful”, “Somewhat useful” or “Not Useful”. Statements included research around: (i) information (ii) policy, (iii) communication, (iv) remote monitoring, (v) mobility, (vi) new ways of learning, (vii) voice activated technologies, (viii) finance based and (ix) VR/AR. VR = Virtual Reality, AR= Augmented Reality.

**Figure 3 ijerph-18-12291-f003:**
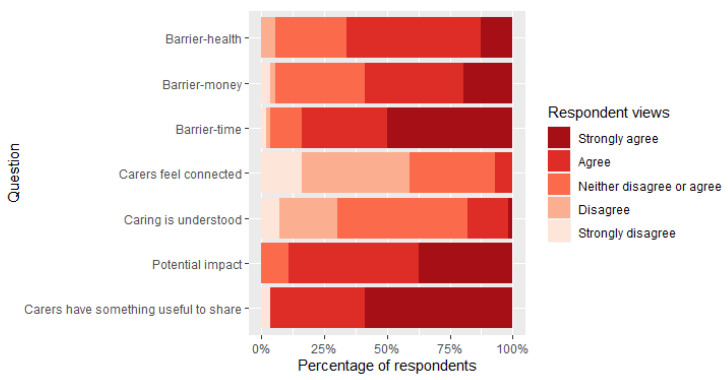
Professionals’ responses to a range of statements according to whether they “Strongly agree”, “Agree”, “Neither disagree or agree”, “Disagree” and “Strongly Agree”. Statements focused on whether: (i) caregivers have something useful to share with researchers, (ii) researchers could make potential impact on caregiver health and wellbeing, (iii) the role of a caregiver is understood, and (iv) whether caregivers feel connected to researchers at university settings. In addition, we explored views on the barriers of (v) time, (vi) money and (vii) health to participation in research.

**Figure 4 ijerph-18-12291-f004:**
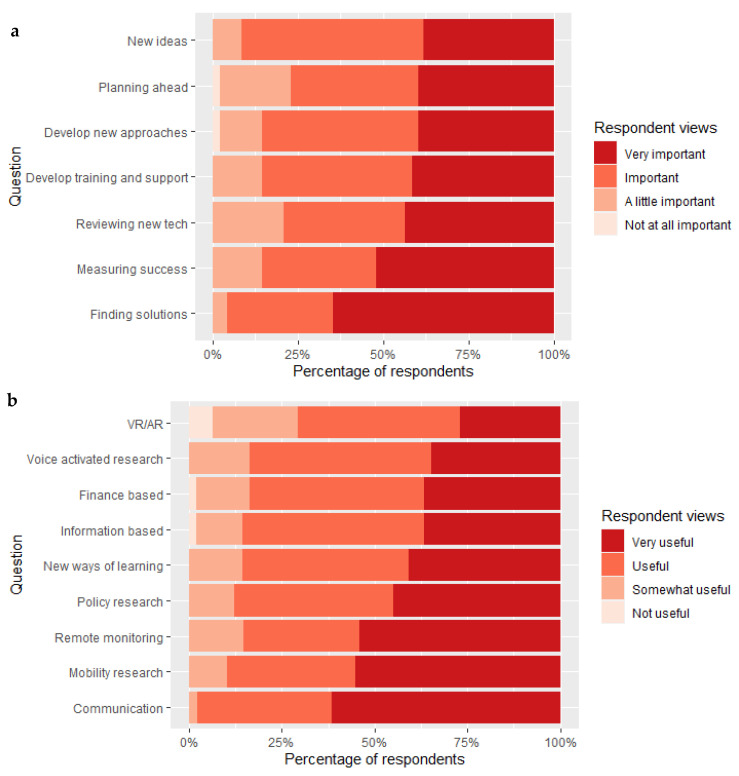
(**a**) Professionals’ responses to a range of statements according to whether they are “Very important”, “Important”, “A little important”, “Not at all important”. Statements included whether caregivers were interested in (i) finding solutions, (ii) developing new approaches to work with researchers, (iii) measuring success, (iv) developing training and support, (v) planning ahead, (vi) developing new ideas, and (vii) reviewing new technologies. (**b**) A separate question for professionals focused on whether future work would be “Very useful”, “Useful”, “Somewhat useful” or “Not Useful”. Statements included research around: (i) information (ii) policy, (iii) communication, (iv) remote monitoring, (v) mobility, (vi) new ways of learning, (vii) voice activated technologies, (viii) finance based and (ix) VR/AR.

**Figure 5 ijerph-18-12291-f005:**
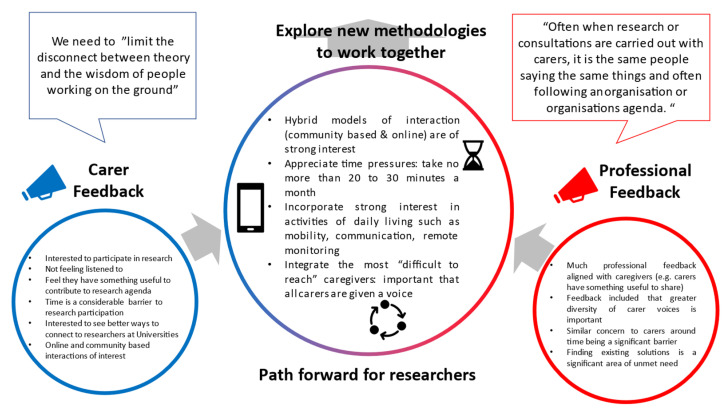
Summary figure summating both informal caregiver and professionals/innovators interested in the caregiver demographic.

**Table 1 ijerph-18-12291-t001:** Participant Demographics. * n = 61 for professionals’ gender question.

Variable	Group	Caregivers (n= 112, [%])	Professionals (n = 62 *, [%])
Age Group	18 to 24	0 [0%]	0 [0%]
	25 to 34	4 [3.6%]	16 [26%]
	35 to 44	21 [18.8%]	13 [21%]
	45 to 54	47 [42%]	22 [35%]
	55 to 64	30 [26.8%]	11 [18%]
	65 to 74	7 [6.3%]	0 [0%]
	75 to 84	2 [1.8%]	0 [0%]
	Prefer not to say	1 [0.9%]	0 [0%]
Gender	Man/Male (including trans man)	12 [10.7%]	13 [21%]
	Woman/Female (including trans woman)	96 [85.7%]	48 [79%]
	In another way	2 [1.8%]	0 [0%]
	I prefer not to answer	2 [1.8%]	0 [0%]
Ethnicity	White	111 [99.1%]	58 [94%]
	Asian/Asian British	0	2 [3%]
	Other ethnic group	0	2 [3%]
	Mixed/multiple ethnic groups	1 [0.9%]	0 [0%]
Education level	Degree or equivalent	68 [60.7%]	46 [74%]
	Higher education	27 [24.1%]	10 [16%]
	Other qualifications	5 [4.5%]	5 [8%]
	School qualifications	10 [8.9%]	1 [2%]
	No qualifications	2 [1.8%]	0 [0%]

**Table 2 ijerph-18-12291-t002:** Key themes and sub-themes from Caregiver Survey. (CAHMS= Child and Adolescent Mental Health Services).

Key Theme	Sub-Theme	Example Quote (s)
1. Previous research participation	1.1. Experience and connection to research/ers	* **The research facilitated access to [a] psychologist and assessment whilst [I was] awaiting CAHMS appointments** *
	1.2. Barriers	* **I always found great difficulty finding the time with all other responsibilities, however, feel it would have been good opportunity, but struggle with other commitments to find time** *
2. Future research participation	2.1. Value of the input from caregivers	* **It is important to have your voice heard to give an accurate picture of caring and carers** *
	2.2. Methods of participation	* **Face-to-face is always preferred to build meaningful relationship. Online is next best thing and personally I would be reluctant for the final two options [telephone and post] as [it’s] difficult to engage** *
	2.3. Time available for participation	* **There needs to be a balance between the time commitment and the formation of a relationship between the carer and the researcher** *
3. Future research aspirations	3.1. Innovative technology	* **The use of remote technology so that carers don’t worry or have to be with the person as much** *
	3.2. Improved support for carers	* **Give carers a voice and some real support, there are many of us who are unpaid and dedicate** * * **ourselves to our person whilst struggling with life ourselves** * * **Please help us be less invisible in our communities. Please help us help the person we love and care for to be less invisible in our communities.** *
	3.3. Impact on policy	* **Shaping government agenda is probably where change is mostly required** *

**Table 3 ijerph-18-12291-t003:** Key themes and sub-themes from Professionals Survey.

Key Theme	Sub-Theme	Example Quote
1. Previous research participation	1.1. Experience and connection to Research/ers	* **I feel obligated to take part since we need more attention on the work we do as carers.** *
	1.2. Interest in research	** *Think it is really important to gather information from all walks of life and people who are doing the work at ground level and not people sitting in offices that are not meeting clients and families on a daily basis.* **
	1.3. Barriers	** *Time is my main commodity.* **
2. Future research participation	2.1. Methods of participation	** *Carers have little time for themselves, so it would make sense to provide ways in which they could contribute with their experiences at a convenient time. They are exhausted day by day so asking them to go somewhere or receiving people at home is uncomfortable and burdensome for them.* **
	2.2. Involving hard to reach stakeholders	** *Often when research or consultations are carried out with carers, it is the same people saying the same things and often following an organisation or organisation’s agenda. Would suggest there is value in speaking with carers who haven’t been supported by carer organisations.* **
3. Future research aspirations	3.1. Innovative technology	** *Technology for monitoring care for persons when they have memory issues that can give the carer peace of mind…Having a life outside of caring, maybe tracking what the carer is doing and encouraging them to take a break where possible.* **
	3.2. Training and support for Carers	* **Training and peer support is currently a massive challenge and services are currently looking to move as much as possible onto online platforms.** * * **To find the best possible solutions to help people manage their own well-being as carers, which will impact on those they care for.** *

## Data Availability

The data presented in this study are available on request from the corresponding author. The full data are not publicly available to ensure anonymity of participants who took part in this work.

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
