# Peer review of "Building a Research Roadmap for Caregiver Innovation: Findings from a Multi-Stakeholder Consultation and Evaluation"

_ijerph, 2021, doi:10.3390/ijerph182312291_

Round 1

Reviewer 1 Report

Please refer to the attached file

Author Response

Response to Reviewer 1 Comments

Many thanks this positive feedback and constructive advice.

The main recommended 
changes are 1. Eliminate non-structural abstracts. For example, in the summary's 
contents, the titles of Background, Methods, and Results might be considered 
cancelled.

Many thanks for this useful feedback - we have followed this suggestion for the abstract.

2. Write specific conclusions: This section is mandatory and briefly

Again, thanks for this useful feedback - we have followed this suggestion and highlighted this section with an appropriate heading.

Reviewer 2 Report

Dear Authors,

Please kindly accept my comments and revisions below, which are formulated constructively to improve the quality of your manuscript.

The authors evaluated caregiver engagement by undertaking a multi-stakeholder consultation via a survey (with 174 responses) to design a future roadmap for innovation in caregiver research. The manuscript ‘Building a research roadmap for caregiver innovation: findings from a multi-stakeholder consultation and evaluation’ fits the aims and scope of the International Journal of Environmental Research and Public Health, and it is adequate for publication.

Lines 11-33: In the abstract, remove the words ‘background’ (line 11), ‘methods’ (line 17), ‘results’ (line 22) and ‘conclusions’ (line 27) and re-write the sentences.

Line 43: Remove “This story” as it is inappropriate in a scientific context and choose different terminology.

Line 74: WMO (World Health Organization).

Lines 95-97: Please explain why the surveys are different for caregivers and professionals. Mention the number of questions that are different. Same comment to lines 289-290.

Lines 96 and 105: Please give examples of the “professional population”.

Lines 91 - 143: Material and Methods.

The geographical location is currently omitted, and this information is vital. Was the survey aimed at caregivers and professional groups in all UK? Please indicate. A total of 174 responses are a small sample of the UK caregivers and professional population, and it should be stated as a limitation of this research.

Another weak point of this study is that no information about the geographical location of the participants was collected during the survey. It would be interesting to analyse the responses taking into consideration rural vs urban and different regions in the UK and taking into consideration diverse socio-economic backgrounds. This limitation should be stated in the manuscript and perhaps should be considered in future research.

Lines 119-122: Provide examples of social media channels. Define “email distribution through networks”. This information is essential to assess whether representative samples of the populations of caregivers and professionals are included in this research. This section requires more details on the channels used to reach participants.

Lines 124-134: Provide more insight and references on the qualitative data analysis.

Line 174: Table 2 – define CAHMS between brackets. This information is important for readers outside the UK.

Lines 200 (figure 1), 240 (figure2a), 272 (figure 3), 336 (figure 5) require higher resolution figures for better reading.

Lines 144-314 (results section): As an overall comment, the sentences in this section should include more information on the percentage of specific responses. In parts, the graphs are cited; however, the argumentation should be supported by a more detailed description of percentages in the text. Examples include lines 181 (“most”), 222 (“many”), 286 (“many respondents”), which provide no information. Please add the percentages.

Lines 227-229: The word support is repeated four times. Re-organise the sentence.

Line 272: Responses, not repsones.

Lines 280-283: Re-organise the sentence as it is too long and hard to read.

Lines 295-297: Re-organise the sentence as it is not clear.

Line 306: Figure 4 is missing.

Line 316: Add a reference for “an ageing growing global population.

Line 317: Add a reference, such as a newspaper publication, for “the disruption caused by COVID-19”.

Line 370: Provide a reference for citizen science.
